

# Phylogeny of Cassieae based on seed morphological and ITS evidence

Jing Cai[1,2], Chuang Yang[1], Leyang Wang[1], Jiaqing He[2,†] and Qiang Wang[3]

[1] Hefei Preschool Education College, Hefei, Anhui, China
[2] School of Life Science, Anhui University, Hefei, Anhui, China
[3] National Key Laboratory for Tea Plant Germplasm Innovation and Resource Utilization, Anhui Agricultural University, Hefei, Anhui, China
[†] Deceased.

## ABSTRACT

The Cassieae tribe, comprising economically significant species, is understudied in terms of phylogenetics within China. This study aimed to elucidate the taxonomic status and systematic relationships among Cassieae species by integrating seed coat micro-morphological characteristics with molecular data from the internal transcribed spacer (ITS) region of nuclear ribosomal DNA. (1) The results indicate that the tribe is classified into seven distinct clades; Clade V consists of *Chamaecrista*, which is characterized by primarily monosymmetric flowers with occasional asymmetry, the presence of extrafloral nectaries (EFNs), five or ten stamens, and pods that dehisce elastically. Clade VI is associated with *Senna*, which displays polysymmetric or monosymmetric flowers, uniformly ten stamens or occasionally three staminodes, variable presence of EFNs, and predominantly indehiscent pods, with rare instances of slit dehiscence. Clade VII is characterized by the presence of *Cassia*, which exhibits polysymmetric flowers, consistently ten stamens—three of which are elongated and curved, typically exceeding the length of the petals—and indehiscent pods. (2) The monophyly of *Cassia* and *Senna* was strongly supported, with *Cassia* grouping closely with *Senna*, but distantly from *Chamaecrista*. (3) It is noteworthy that *Senna occidentail* and *Senna sophera* are confirmed as sister taxa, while *Senna corymbosa* and *S. bicapsularis* are confirmed as sister taxa. By integrating molecular biology and morphological taxonomy, this research enhances our comprehension of the phylogenetic relationships and evolutionary history within the Cassieae tribe in China.

# INTRODUCTION

The Cassieae tribe, first delineated by Linnaeus in his foundational botanical work "Species Plantarum" (*Carolus, 1753*), has undergone significant taxonomic revisions. In the early 1980s, *Irwin & Barneby (1982)* reclassified Cassieae into five subtribes, a classification that since been superseded by modern systems. Current taxonomic frameworks, including the *Azani et al. (2017)*, *Ringelberg et al. (2022)*, and *Rando et al. (2024)*, recognize Cassieae as a tribe that includes not only the genera *Senna*, *Cassia*, and *Chamaecrista* but also *Batesia*, *Melanoxylum*, *Recordoxylon,* and *Vouacapoua*. Globally, Cassieae encompasses over 600 species, with approximately 20 species documented in China by *Wu et al. (2010)*. The

Corresponding author
Qiang Wang, wqiang@ahau.edu.cn

majority of these species, both wild and introduced, belong to the genera *Senna* and *Cassia*, exhibiting diverse ecological forms ranging from herbs to shrubs, subshrubs, and trees. Botanical references such as "China Flora" (*Chinese Academy of Sciences, 1988*), "Flora de Chile" (*Reiche, 1898*), and "Flora of Pakistan" have adhered to Linnaeus's framework, reflecting a lag in the revision of the Leguminosae family. This has resulted in a disjointed classification of Cassieae, and reports on its phylogeny in China being scarce and its nomenclature often confusing.

Modern taxonomy within Cassieae prioritizes floral morphology as a key to evolutionary patterns, focusing on stamen configuration, corolla symmetry, and floral type, diverging from traditional taxonomies that relied heavily on seed and fruit structures. A distinctive feature of the Cassieae subtribe is enantiostyly, the lateral displacement of styles relative to the floral axis, which enhances floral asymmetry and complements the irregularity of the corolla (*Dulberger, 1981*; *Irwin & Barneby, 1982*; *Gottsberger & Silberbauer-Gottsberger, 1988*). Additionally, the presence of extrafloral nectaries (EFNs) on petioles or pedicels is a significant characteristic, playing a crucial role in plant defense by attracting ants and deterring herbivores (*Heil & Mckey, 2003*). Despite their taxonomic importance, the genetic basis and evolutionary implications of EFN distribution and structural attributes remain underexplored. Since the 1950s, seed micromorphological characteristics have been applied to plant taxonomy and systematics, with studies such as *Benson (1962)*, *Barthlott (1981)*, *Neelam, Muhammad & Mushtaq (2021)*, and *Aysun, Mehmet & Orhan (2024)* demonstrating the taxonomic significance of seed micromorphological structures at various levels, including family, genus, and species. In recent years, seed coat micromorphology has been employed as an auxiliary method in the classification of legumes (*Chen, Liu & Cai, 2007*; *Li, Liu & Chen, 2008*; *Neelam, Muhammad & Mushtaq, 2021*), yet there is a dearth of studies on Cassieae in China utilizing this approach.

This study employed nrDNA internal transcribed spacer (ITS) sequences from 22 Cassieae species, with three outgroup taxa, to reconstruct the evolutionary tree using both maximum likelihood (ML) and Bayesian methods. The phylogenetic analysis complemented by seed micromorphology, flower morphology, and EFN traits. This integrated approach allows for a comprehensive analysis of character evolution, aids in defining generic boundaries within Cassieae, clarifies species limits, and elucidates the phylogenetic relationships among *Cassia*, *Chamaecrista*, and *Senna* in China. Consequently, this research aligns Chinese classification research on Cassieae with international standards, laying a foundational classification framework for future studies on Chinese medicinal materials, including those of the Cassieae tribe.

## MATERIALS AND METHODS

The Cassieae tribe is widely distributed across various climatic zones, including tropical, subtropical, and temperate regions, with a global presence in the Americas, Asia, Africa, and Oceania. Within China, this subtribe is predominantly found in provinces and territories situated south of the Yangtze River, as documented by *Wu et al. (2010)*. Our study collected samples from 10 species across Anhui, Hainan, and Guangdong provinces, which are known

**Table 1  The materials used in the experiment and their origin ITS of the ribosomal DNA sequencing.**

|  | Taxa |  | Source | Coordinates | ITS GenBank Acc. No. |
|---|---|---|---|---|---|
|  | *Cassia* | *C. fistula* | Sun Yat-Sen University | 23°09′N, 113°30′E | GU175310 |
|  | *Chamaecrista* | *Chamaecrista nomame* | Hefei Dongpu Reservoir | 31°52′N, 117°12′E | HQ833046 |
| Cassieae |  | *S. tora* | Anhui University Experimental Garden | 31°84′N, 117°25′E | FJ572046 |
|  |  | *S. bicapsularis* | Anhui University Experimental Garden | 31°84′N, 117°25′E | HQ833043 |
|  |  | *S. alata* | Anhui University Experimental Garden | 31°84′N, 117°25′E | HQ833041 |
|  | *Senna* | *S. didymobotrya* | National Nature Reserve, Hainan Jianfengling | 18°25′N, 108°47′E | HQ833045 |
|  |  | *S. occidentalis* | Anhui Suzhou | 34°24′N, 116°21′E | HQ833044 |
|  |  | *S. sophera* | Anhui University Experimental Garden | 31°84′N, 117°25′E | HQ833042 |
|  |  | *S. corymbosa* | University of Science and Technology of China | 31°83′N, 117°26′E | HQ833047 |
|  |  | *S. siamea* | South China Normal University | 23°05′N, 113°38′E | – |

for their rich biodiversity and representativeness of the tribe's distribution in China. Details regarding the collection sites and origins of the samples, along with the ribosomal DNA internal transcribed spacer (ITS) sequences, are presented in Table 1.

## Seed morphology and seed-coat micromorphology observation

The mature, intact seeds from Cassieae representatives were examined under light microscopy to document and illustrate their external features, with a meticulous measurement of their linear dimensions: length, width, and thickness. For microscopic inspection, seeds were preprocessed by immersion in 75% ethanol for sterilization, followed by ultrasonic cleaning to remove superficial contaminants. Subsequently, they were dried at 70 °C for 30 min and coated with gold for 90 s for scanning electron microscopy (SEM) imaging, using a HITACHI S-4800 instrument. The seed-coat micromorphology was described using the terminology standardized by *Liu, He & Lin (2004)*. Optical micrographs and SEM images were presented in Figs. 1 and 2.

## ITS gene sequencing

Total genomic DNA extraction from silica gel-preserved leaf samples was conducted using the cetyl trimethyl ammonium bromide (CTAB) protocol described by *Doyle & Doyle (1990)*. The ribosomal DNA ITS region was amplified *via* polymerase chain reaction (PCR) utilizing the universal primers ITS4 (forward: 5′-TCCTCCGCTTATTGATATGC-3′) and ITS5 (reverse: 5′-GGAAGTAAAAGTCGTAACAAGG-3′), as outlined by *White et al. (1990)*. The PCR mixture comprised a total volume of 30 µL, including 3 µL of 10× PCR buffer, 0.6 µL of 10 mM dNTP mix, 0.5 µL each of forward and reverse primers, 0.5 µL of Taq DNA polymerase, and approximately 50 ng of total DNA. The thermal cycling profile entailed an initial denaturation at 95 °C for 4 min, followed by 40 cycles of 95 °C for 1 min, 52 °C for 1 min for annealing, and 72 °C for 1 min for extension, concluding with a final extension at 72 °C for 10 min. Amplified PCR products were then submitted to Sangon Biotech (Shanghai) Co., Ltd., for purification and sequencing. In order to ascertain sequence accuracy, bidirectional sequencing was performed on both the sense and antisense

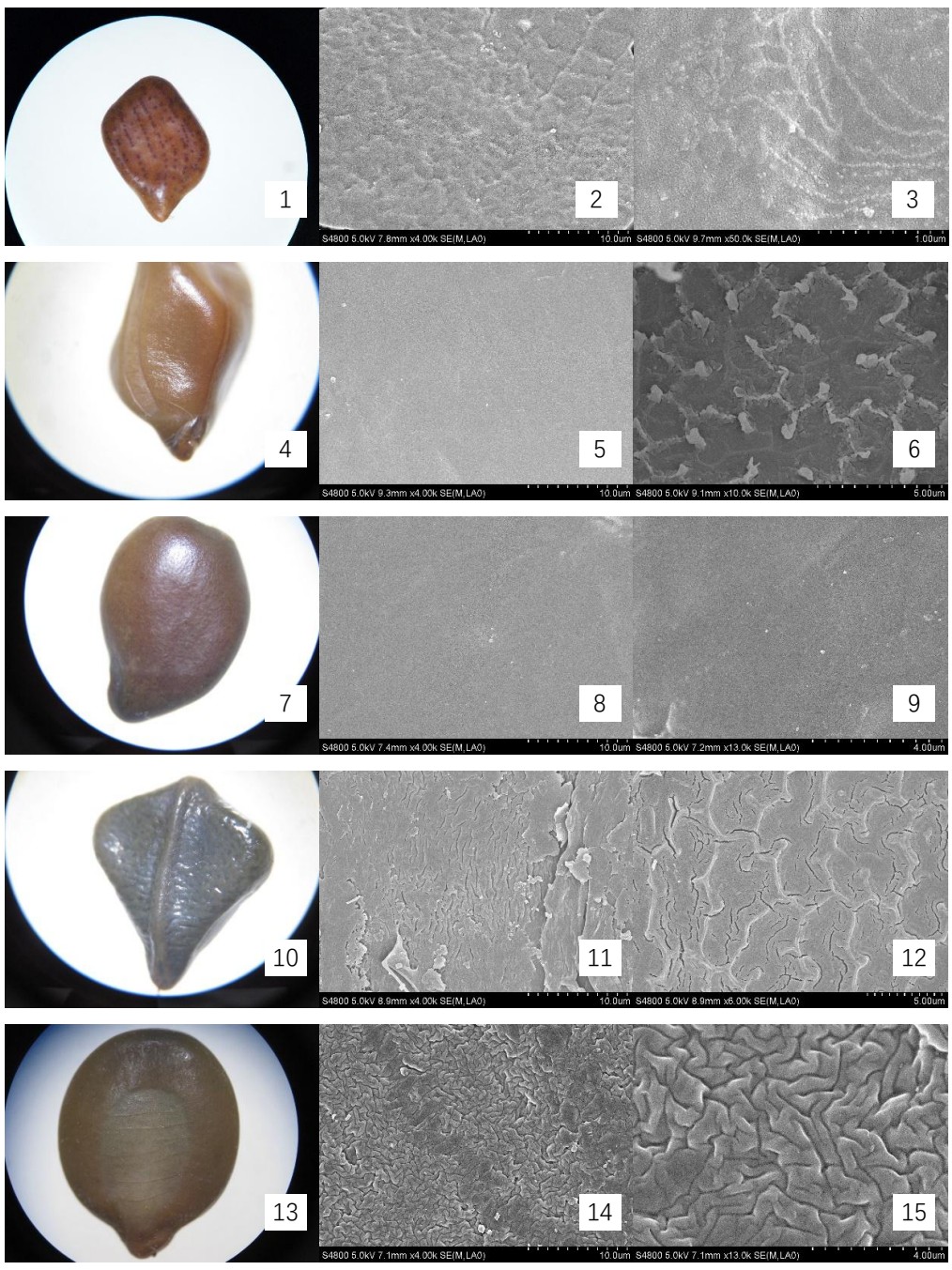

**Figure 1** **Micromorphology of seed coat of Cassiinae is and related genera.** (1–3) *Chamaecrista nomame* (×30;×4k;×50k); (4–6). *S. tora* (×30;×4k; Palisade tissue, ×50k); (7–9). *S. bicapsularis* (×30;×4k;×13k); (10–12). *S. alata* (×25;×4k;Palisade tissue, ×6k); (13–15). *S. didymobotrya* (×30;×4k;×13k).

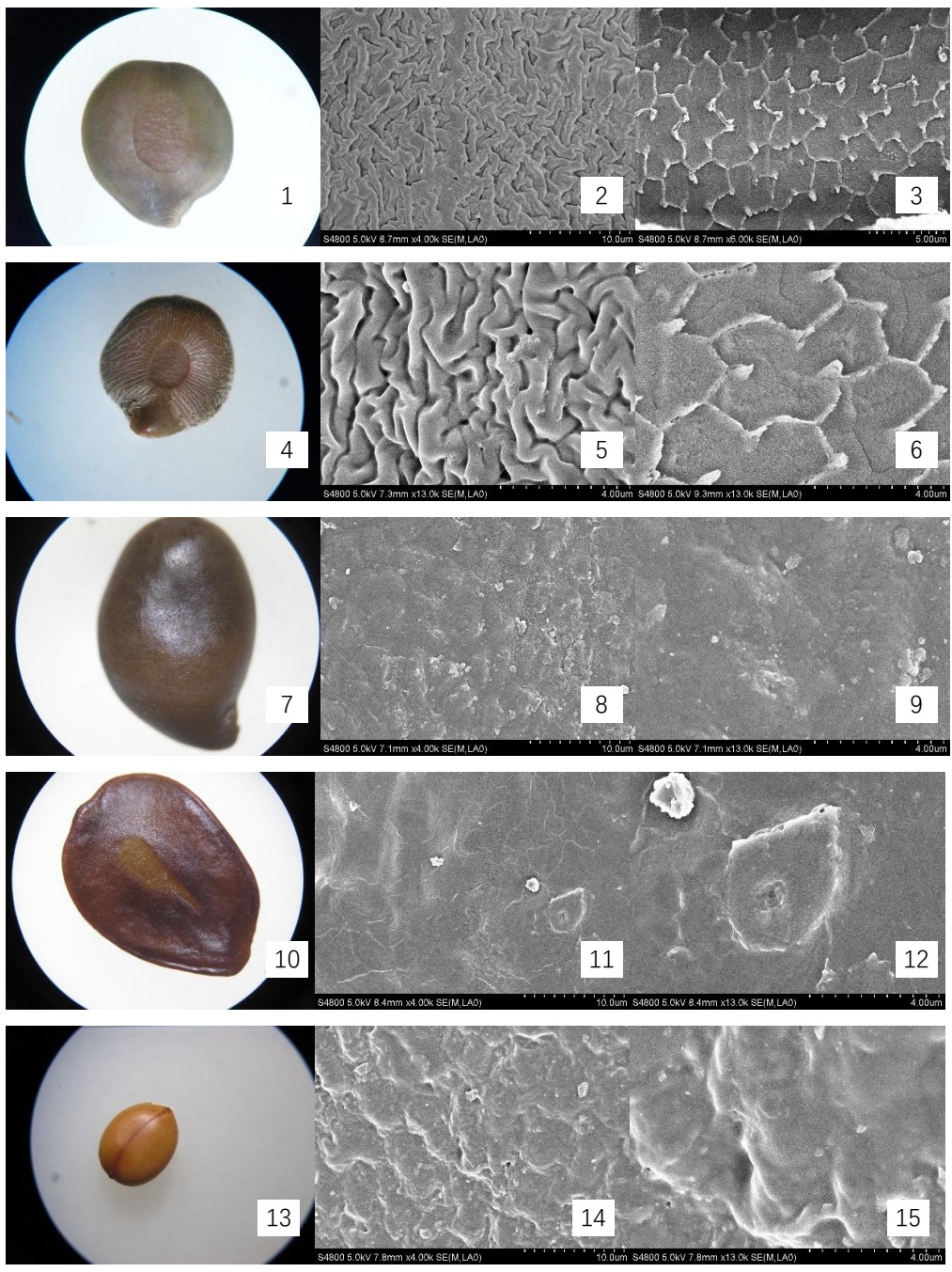

**Figure 2** **Micromorphology of seed coat of Cassiinae is and related genera.** (1–3) *S. occidentalis* (×30;×4k;Palisade tissue, ×6k); (4–6). *S. sophera* (×30;×13k;Palisade tissue, ×13k); (7–9). *S. corymbose* (×30;×4k;×13k); (10–12). *S. siamea* (×25;×4k;×13k); (13–15). *Cassia fistula* (×10;×4k;×13).

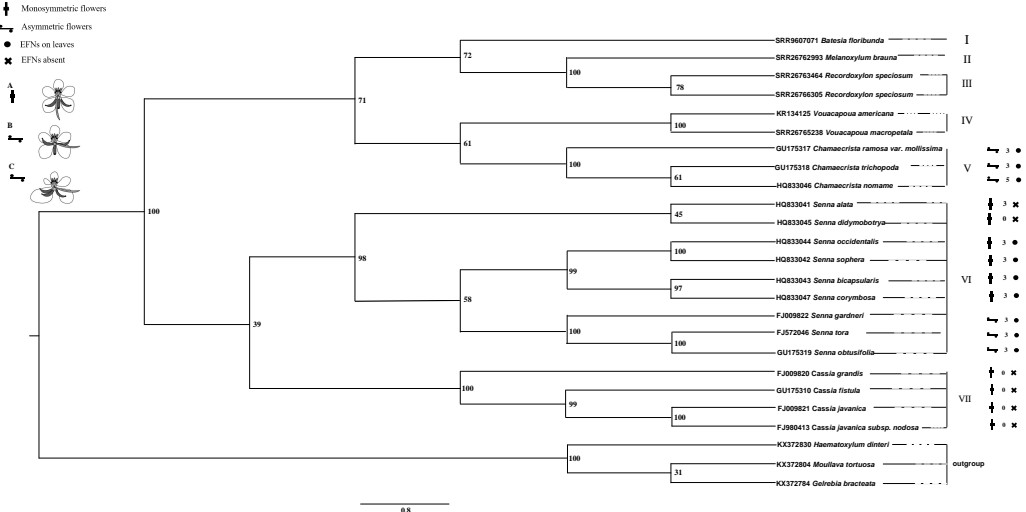

**Figure 3** **The phylogenetic tree of Cassieae based on ITS nrDNA sequences.** Number above the branches are bootstrap values greater than 50% frequencies (1,000 replications). Numbers above branches show bootstrap value from Maximum Likelihood method. (A) Monosymmetric flower; AL, abaxial lateral stamen; AM, abaxial median stamen; G, gynoecium; LP, lower petal; MI, set of four middle stamens; ST, set of three adaxial staminodes; UP, upper petal; sepals not shown. (B) Asymmetric flower in which only the gynoecium is involved in the floral asymmetry. (C) Asymmetric flower in which also petals and stamens are involved in the floral asymmetry.

strands for the ITS region of every species. ITS sequences of seven out of ten species were sequenced and subsequently deposited in GenBank (Table 1).

## Phylogenetic analysis

The ITS sequences of *Batesia floribunda*, *Melanoxylum brauna*, *Recordoxylon speciosum* and *Vouacapoua macropetala* were found using GeneMiner v.1 (*Xie et al., 2024*) after being concatenated from the SRA data (accession nos. are SRR9607071, SRR26762993, SRR26763464/SRR26766305 and SRR26765238). For the remaining species, sequences were retrieved from GenBank databases. The outgroup taxa comprised *Haematoxylum dinteri*, *Moullava tortuosa* and *Gelrebia bracteata*. Sequence alignment and editing were performed using the SEQMAN software, followed by assembly in MEGA v. 11. Maximum likelihood phylogenetic trees were conducted in RAxML v. 8 (*Alexandros, 2014*). The ITS sequence matrix's optimal nucleotide substitution model was determined to be GTR+I+G by using MrModel test v.3.06 software, and the maximum likelihood (ML) analysis was performed with 1,000 iterations, initiated from a randomly generated tree and selected the best score tree supported by the bootstrap score value (Fig. 3). Bayesian phylogenetic trees were conducted in Mrbayes (*Huelsenbeck, Ronquist & Nielsen, 2001*). The optimal nucleotide substitution model was calculated GTR+I+G using jModelTest.

## RESULTS

### Microscopic observation of seed coats

The seeds of the ten Cassieae species exhibited a variety of shapes, primarily elliptical, broadly ovate, or rhombic, alongside significant differences in size. *Chamaecrista nomame* had the smallest seeds, with dimensions averaging approximately $(3.67 \pm 0.18) \times (3.66 \pm 0.14) \times (0.98 \pm 0.13)$ mm, while *Cassia fistula* had the largest, measuring around $(9.25 \pm 0.63) \times (7.12 \pm 0.12) \times (3.64 \pm 0.37)$ mm. The coloration of the seeds spanned from yellowish-brown to green-brown and brown hues, typically with a small diminutive hilum located at the apex or its sides. Variations were observed in pod dehiscence patterns: *Cassia* was characterized by indehiscent pods, which may represent an ancestral trait; *Senna* species showed pods that were either indehiscent or split along dorsal/ventral sutures; *Chamaecrista*, on the other hand, had evolved elastic dehiscence, a trait that could be associated with an increased range of seed dispersal. According to *Corner (1976)* study, the size and pattern complexity of seeds were telling of their evolutionary history: larger seeds with simpler sculptures are indicative of an ancestral state, whereas the presence of complex patterns in smaller seeds suggests derived traits. Our microscopic examination corroborated this perspective, revealing a spectrum of seed surface textures, ranging from smooth to micro-grooved or striate under light microscopy. Further scrutiny through SEM magnification unveiled a diverse array of intricate seed coat sculptures, categorized as ruminate, reticulate, undulate, striped, verrucose, favulariate, or smooth. The comparative analysis of seed weight and sculpture among species provides insights into their phylogenetic status. The heavier seeds of *Cassia fistula*, weighing approximately 1.75 g with a favulariate sculpture, aligned with *Corner*'s (*1976*) notion of a primitive condition. In contrast, the lighter seeds of *Chamaecrista nomame*, which weigh around 0.06 g and exhibit complex, ripple-like patterns, were indicative of more recently evolved characteristics. Seeds of intermediate weight, between 0.2 g to 0.35 g, displayed moderate complexity. Notable distinctive patterns were observed among the species: *Senna alata*, *S. corymbosa*, and *S. siamea* exhibited somewhat coarser sculptures; *Senna didymobotrya*, *S. occidentalis*, and *S. sophera*, all with ruminate textures, varied in edge width; *S. tora* and *S. bicapsularis* had smooth coats, signifying a close relationship. These findings indicated that Chamaecrista, Cassia, and Senna probably represented independently monophyletic lineages.

During the examination, it was observed that the seed coat cuticles of *S. tora*, *S. occidentalis*, *S. sophera*, and *S. alata* anomalously split, revealing a uniform reticulate exine pattern with palisade tissue cells underneath. The first three species exhibited linear murus swelling, while *S. alata* displayed arcuate swelling. This phenomenon, not observed in other species under similar conditions, suggests a possible correlation with differences in cuticle structure and thickness. Comprehensive descriptions of the macroscopic seed characteristics and seed coat micromorphology for all ten species were detailed, with a summary provided in Table 2 and visual representations in Figs. 1 and 2.

**Table 2** **Seed morphology and seed coat micromorpholog in Cassieae.**

| Taxon | Shape of Legume | Size of seed (L×W×T) (mm) | Shape of seed | Colour of seed | Micromorphological features of seed coat | Plate |
|---|---|---|---|---|---|---|
| *Chamaecrista nomame* | Legume compressed, hairy 3–8 × ca. 0.5 cm. Elastically dehiscent. | (3.67 ± 0.18) × (3.66 ± 0.14) ×(0.98 ± 0.13) | Seeds 6–12, compressed, subrhomboid. | yellowish-brown, smooth. | Reticulate, accompanied by ripples same as surface specks shape | Fig. 1-1, 2, 3 |
| *S. tora* | Legume terete, subtetragonous, slender, 10–15 × 0.3–0.5 cm, both ends acuminate, valves membranous. Indehiscent. | (0.23 ± 0.01) × (4.57 ± 0.09) × (2.5 ± 0.14) | Seeds 20–30, rhomboid, with an areole. | green-brown, glossy. | Smooth; palisade tissue is double reticulate. | Fig. 1-4, 5, 6 |
| *S. bicapsularis* | Legume brown, terete, straight or slightly curved, ca. 15 cm, valves membranous. Indehiscent. | (5.72 ± 0.18) × (3.79 ± 0.18) × (1.81 ± 0.05) | Seeds 50–60, ovoid, flattened. | brown | Smooth; a spot of undulate | Fig. 1-7, 8, 9 |
| *S. alata* | Legume winged, sharply tetragonal, 10–20 × 1.5–2 cm, glabrous, with a broad, membranous wing down middle of each valve. Indehiscent. | (6.08 ± 0.17) × (5.11 ± 0.12) × (1.57 ± 0.15) | Seeds 50–60, compressed, deltoid. | green-brown, glossy. | Undulate; palisade tissue is fine-reticulate, ridge of reticu lum is anomalous polygon. | Fig. 1-10, 11, 12 |
| *S. didymobotrya* | Ligulate-oblong, 8–10 cm × 1–1.8 cm, leathery, bicarinate by sutures, apex with a long and slender awn. Indehiscent. | (5.67 ± 0.01) × (4.46 ± 0.14) × (1.67 ± 0.04) | Seeds 9–16, obovoid-oblong, flattened | yellowish-brown | Ruminate, fine-ridge, ca. 0.4 μ m. | Fig. 1-13, 14, 15 |
| *S. occidentalis* | Legume brown, with pale thick margins, strap-shaped, falcate, flattened, 10–13 × ca. 1 cm, with septa between seeds. Indehiscent. | (4.35 ± 0.07) × (3.9 ± 0.28) × (1.87 ± 0.02) | Seeds 30–40, flat, orbicular, 3–4 mm in diam. | green-brown and brown. smooth. | Ruminate, wide-ridge, ca. 1 μ m; palisade tissue is fine-reticulate, ridge of reticu lum is anomalous polygon, ca. 2 μ m | Fig. 2-1, 2, 3 |
| *S. sophera* | Legume straight, 5–10 × 0.5–1 cm, flattened and slightly thick at first, subcylindric, ± swollen when ripe. Indehiscent. | (4.95 ± 0.12) × (4.31 ± 0.18) × (1.97 ± 0.17) | Seeds 30–40, ovoid, compressed. | green-brown and brown. | Ruminate, fine-ridge, ca. 0.5 μ m; palisade tissue is fine-reticulate | Fig. 2-4, 5, 6 |
| *S. corymbosa* | terete, ca. 5–8 cm. Indehiscent. | (5.95 ± 0.09) × (3.96 ± 0.37) × (2.67 ± 0.12) | Seeds 30–40, ovoid, | brown. | verrucous | Fig. 2-7, 8, 9 |
| *S. siamea* | Legume flattened, 15–30 × 1–1.5 cm, suture thick, riblike, pubescent, purplish brown when mature. Indehiscent. | (7.54 ± 0.19) × (5.69 ± 0.25) × (0.75 ± 0.04) | Seeds 10–30, light brown, ovoid. | brown. | Stripe | Fig. 2-10, 11, 12 |
| *Cassia fistula* | Legume pendulous, blackish brown, terete, sausage-shaped, 30–60 cm × 2–2.5 cm in diam. Indehiscent. | (9.25 ± 0.63) × (7.12 ± 0.12) × (3.64 ± 0.37) | Seeds numerous, separated by papery septa, glossy brown, elliptic, flattened. | yellowish-brown | Favulariate, with Foveate and Granu late | Fig. 2-13, 14, 15 |

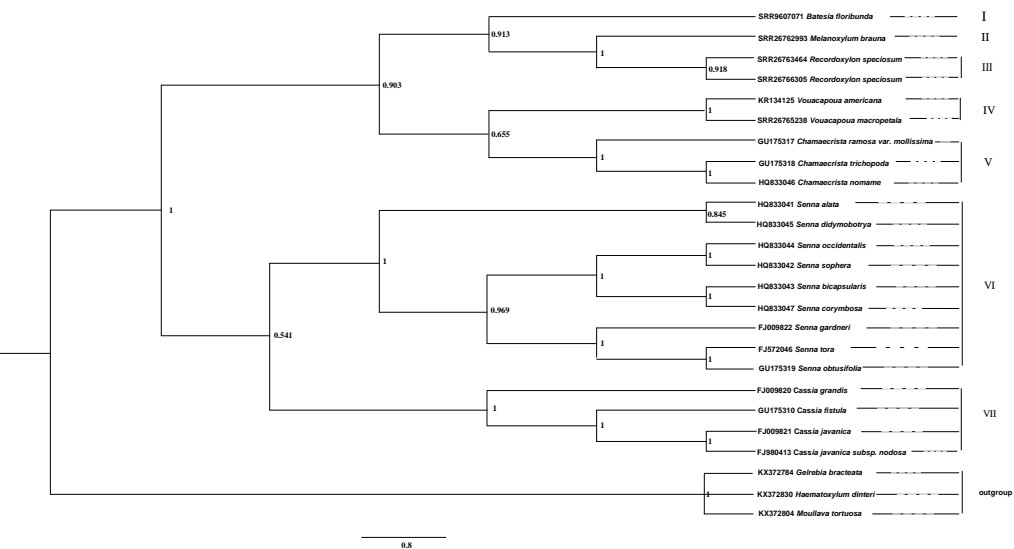

**Figure 4** The Bayesian phylogenetic tree of Cassieae based on ITS nrDNA sequences.

## Phylogeney based on ITS sequence

The phylogenetic trees derived from Bayesian and maximum likelihood (ML) analyses showed remarkable congruence, the topological structure of Cassieae can be divided into seven main groups (Figs. 3 and 4). There are two branches formed: *Cassia*, and *Senna* were revealed as monophyletic groups, and *Chamaecrista* is more closely related to *Batesia*, *Melanoxylum*, *Vouacapoua* and *Recordoxylon*. Expanding on *Brigitte, Peter & Luciano (2006)* work with Senna's floral diagram, this study developed a comprehensive floral diagram for *Chamaecrista, Cassia* and *Senna* that includes annotations of stamen degeneration patterns. The findings confirmed that these plants typically possesses 10 stamens, with the three abaxial stamens featuring sigmoidally curved filaments that often surpass the anther length. Specifically, Clade VII, representing *Cassia*, maintained this stamen count and morphology, presenting a monosymmetric flower (Type A). Clade VI, corresponding to *Senna*, showed variations: *Senna didymobotrya* exhibits 10 stamens without extrafloral nectaries (EFNs), indicative of an ancestral condition (Type A). *Senna tora*, *Senna obtusifolia*, *Senna gardneri*, *Senna bicapsularis*, *Senna corymbosa*, *Senna occidentalis* and *Senna sophera*, with EFNs, exhibited a reduction to three stamens, marking secondary evolutionary development, but differed in stamen types between the clades (Type B for *Senna gardneri*, *Senna tora*, *Senna obtusifolia* and Type A for *Senna Occidcidae*, *Senna sophera*, *Senna corymbosa*, *Senna bicapsularis*). Finally, Clade V, characterized by *Chamaecrista*, displayed monosymmetric to asymmetric flowers with stamen type C and the presence of EFNs, highlighting the divergent evolutionary paths within Cassieae.

## DISCUSSION

While China hosts a relatively modest number of Cassieae species—approximately 20—these plants have been integral to traditional Chinese medicine for centuries. The
"Sheng Nong's Herbal Classic" extols the virtues of *Senna tora*, noting its long-term use can "strengthen the kidneys, brighten the eyes, and alleviate constipation" (*National Pharmacopoeia Committee, 2005*). Despite their medicinal use, studies on these species are lacking, and there is considerable taxonomic confusion, particularly at the tribe, genus, and species levels, with the same species often bearing different latin names. This taxonomic ambiguity hampers the full realization of their economic potential, as these plants are not only used in traditional medicine, but also as forage and ornamentals.

ITS-based phylogenetic analysis conducted in this study corroborates the classification of Cassieae into seven principal clusters, aligning with recent international research (*Azani et al., 2017*; *Ringelberg et al., 2022*; *Rando et al., 2024*). The phylogenetic position of *Vouacapoua* is controversial and difficult, while the other six genera have a clear taxonomic position. Initially, *Haston, Lewis & Hawkins (2005)* reported non-monophyly within the genus, based on sampling from only two species. The matK analysis of *Azani et al. (2017)* included all three species of the genus *Vouacapoua*, supporting the genus as monophyletic, and in all phylogenetic analyses, *Vouacapoua* was resolved as part of the Cassieae clades, or as a sister group of the Cassieae clades. The finding is corroborated by subsequent studies including those by *Bruneau et al. (2008)*, *Manzanilla & Bruneau (2012)*, *Azani et al. (2017)*, and *Kates et al. (2024)*. Additionally, our analysis of ITS sequence data from SRA supports the hypothesis that *Vouacapoua* is the sister group to *Chamaecrista*, adding another layer to the intricate phylogenetic network within the Cassieae. Tucker's study (*1996*) supports the separation of the three genera within the *Cassia* genus based on differences in individual flower development. In all the species studied, the order of floral organ initiation is sepals, petals, apetalous stamens plus carpels, and finally apetalous stamens. The sepals of all three genera start in a spiral shape, however, whether the first sepal starts at the mid dorsal position (*Senna*), or at the mid dorsal and off central positions (*Cassia Javanica*), is a rare characteristic state in legumes. The order of petal initiation is different: *Senna*'s is spiral shape, and *Cassia* and *Chamaecrista*'s is unidirectional shape. The rotation of two stamens is consistent and unidirectional. Individual differences between genera occur in leaf sequence, inflorescence structure, formation of small bracts, overlapping initiation between organ whorls (calyx/corolla of *Cassia* seed; two stamen whorls of *Chamaecrista*), eccentric initiation on one side of the flower, anther attachment, anther pore structure, and early mature carpel initiation of *Senna*. The asymmetric corolla and male flowers of *Chamaecrista* are caused by the initiation of early maturing organs on one side (left or right). There is convergence among the three genera, based on inter genus differences in early floral ontogenesis (position of flowers in inflorescence, presence of bracts, position of first sepal initiation, order of petal initiation, asymmetric initiation, overlap between whorls, anther morphology, and timing of carpel initiation), causing similarity during flowering (bright, mostly yellow tray shaped flowers, heteromorphic stamens, sporicidal anther dehiscence, bee pollination, and stigma). From the phylogenetic tree, the monophyly of *Cassia* and *Senna* was strongly supported, with *Cassia* grouping closely with *Senna*, but distantly from *Chamaecrista*. From the seed micromorphology, *Cassia* has large seeds and simple carvings, which shows the characteristics of ancestors. *Chamaecrista* has relatively small seeds that exhibit derivative characteristics. The seeds of *Senna* were of medium

weight and showed different complexity. *Senna gardneri*, *Senna tora*, *Senna obtusifolia* has a relatively advanced seed-coat microform. Although *Senna Occidcidae*, *Senna sophera*, *Senna corymbosa* and *Senna bicapsularis* had primitive seed coat micromorphology, the phylogenetic tree further found that: *Senna occidentail* and *Senna sophera* are confirmed as sister taxa, while *Senna corymbosa* and *Senna bicapsularis* are confirmed as sister taxa.

*Senna corymbosa* was aboriginal to Argentina in South America, had been domesticated and cultivated widely. Its introduction was started in 1990s, research data about it was rear. This study analyzed its seed coats micromorphology, classification and relationship. Its pod is similar to a cylinder, 5–8 cm long. Oval seeds often narrow at one end, 5.6–6 mm in length, 2.5–4.2 mm breadth, and about 2.5 mm thickness. The seed surface is brown, smooth with small recesses. The microscopic structure is verrucous, together with debris. Seed maturity is Dec. to Feb. of the following year. The phylogenetic tree indicated that the species belonged to genus *Senna*, and was nearer to *Senna bicapsularis* in relationship.

This study, by combining molecular biology and morphological taxonomy, enhances our understanding of the phylogenetic relationships and evolutionary history of Cassieae in China. The integration of micro-morphological and molecular data offers a comprehensive view of the tribe's diversity and evolutionary patterns. Furthermore, elucidating the taxonomic status of Cassieae is not only academically valuable, but also has practical implications for the future utilization of these species in traditional Chinese medicine and horticulture. The taxonomic clarification of *Senna corymbosa* necessitates revisions in botanical documentation, underscoring the enduring relevance of systematic research within this tribe. It is a testament to the dynamic nature of botanical taxonomy and the need for continuous reassessment in light of new evidence. Regrettably, the current study's scope was limited to seed micromorphology and flower structure analysis of three genera—*Cassia*, *Chamaecrista*, and *Senna*—which are widely distributed in China. This limitation resulted in a gap in our dataset, as we were unable to include species from the other four genera of Cassieae: *Batesia*, *Melanoxylum*, *Recordoxylon*, and *Vouacapoua*. Incorporating data on seed morphology and flower structure of these genera would significantly enhance the completeness and robustness of our analysis.

## ACKNOWLEDGEMENTS

We would like to thank Dr. Cai-Fei Zhang (Wuhan Botanical Garden, Chinese Academy of Sciences) for his help in the reconstruction of the phylogenetic tree. Additionally, we are deeply appreciative of Professor Jin Zhong for contribution to the language polishing during the revision process of the manuscript.

### Funding

This study was supported by the Anhui Provincial Department of Education Humanities and Social Sciences Key Research Project (2024AH053138) and Excellent Young Teacher Training Project of Anhui Province (YQYB2024120). The funders had no role in study design, data collection and analysis, decision to publish, or preparation of the manuscript.

## Grant Disclosures

The following grant information was disclosed by the authors:

Anhui Provincial Department of Education Humanities and Social Sciences Key Research Project: 2024AH053138.

Excellent Young Teacher Training Project of Anhui Province: YQYB2024120.

## Competing Interests

The authors declare there are no competing interests.

## Author Contributions

- Jing Cai conceived and designed the experiments, performed the experiments, analyzed the data, prepared figures and/or tables, authored or reviewed drafts of the article, and approved the final draft.
- Chuang Yang performed the experiments, analyzed the data, prepared figures and/or tables, and approved the final draft.
- Leyang Wang performed the experiments, analyzed the data, prepared figures and/or tables, and approved the final draft.
- Jiaqing He conceived and designed the experiments, performed the experiments, analyzed the data, prepared figures and/or tables, authored or reviewed drafts of the article, and approved the final draft.
- Qiang Wang conceived and designed the experiments, performed the experiments, analyzed the data, prepared figures and/or tables, authored or reviewed drafts of the article, and approved the final draft.

## Data Availability

The sequences are available in Table 1.

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
