# Peer review of "Phylogeny of Cassieae based on seed morphological and ITS evidence"

_PeerJ, doi:10.7717/peerj.18947_

## Round 0.1 · original submission · Major Revisions

I suggest to rewrite the manuscript with a focus on improving the clarity on the aims and objectives of the study.

·

Basic reporting

The article "Phylogeny of Cassiinae Based on Seed Morphological and ITS evidence" contains information and will be useful to journal readers globally. The study article is properly written and discussed. This work may be accepted after minor revisions

Experimental design

The experimental design is OK. the research work is original primary research within the Aims and Scope of the journal.

Validity of the findings

The article "Phylogeny of Cassiinae Based on Seed Morphological and ITS evidence" contains information and will be useful to journal readers globally. The study article is properly written and discussed. This work may be accepted after minor revisions to the following points:

1. Refine study objectives.
2. Add all abbreviations in full after keywords
3. The discussion is brief. Explain and discuss in greater depth.
4. In Figure 3, all botanical names of the plant names should be italicized.
5. The conclusion is very descriptive and adds brief potential findings.
6. Stick to the journal's reference format for reference.
7. Include DOI numbers in all possible references.
8. Besides the previously given comments, authors are requested to proofread the entire text again to ensure that the amended version is free of any common grammatical problems.

Additional comments

The article "Phylogeny of Cassiinae Based on Seed Morphological and ITS evidence" contains information and will be useful to journal readers globally. The study article is properly written and discussed. This work may be accepted after minor revisions to the following points:

1. Refine study objectives.
2. Add all abbreviations in full after keywords
3. The discussion is brief. Explain and discuss in greater depth.
4. In Figure 3, all botanical names of the plant names should be italicized.
5. The conclusion is very descriptive and adds brief potential findings.
6. Stick to the journal's reference format for reference.
7. Include DOI numbers in all possible references.
8. Besides the previously given comments, authors are requested to proofread the entire text again to ensure that the amended version is free of any common grammatical problems.

Reviewer 2 ·

Basic reporting

1. First of all, please check the language and punctuation. Check species names and correct them related to binomial nomenclature.
2. I suggested that material method part and results part may be checked and methods may be carried to the material methods part from result part.
3. Especially for figures(1 and 2) solutions are too low . Please adjust the solutions at least 600 dpi.

Experimental design

1. The submission should clearly define the research question, which must be relevant and meaningful. The knowledge gap being investigated should be identified, and statements should be made as to how the study contributes to filling that gap.
2. However, I suggested that the material method part and results part may be checked, and methods may be carried to the material method part from the result part.

Validity of the findings

1. For the phylogenetic analyses, how did you decide the GTR+G+I parameter? Did you take any test? For a reliable tree, you must make a test by using the test programs! If you did it please write down how you chose the parameter? (AICs value, etc)
2. In the result part, you said ML and MP analysis showed remarkable congruence, but in the phylogenetic tree, there was only one value. Please indicate both bootstrap values together below each branch, separated with a slash like 95 / 96.
3. Please discuss ITS gene region features in plant phylogeny and especially hybridization events between closer species. Moreover, ITS region doesn’t always reflect morphological separations in systematics but here it does. Please explain these situations in discussion part with new references.
4. Why do you use outgroups from only the Umtiza clade ? If you want to show the relationships of Cassianae species, please add more outgroup from other clades and reanalyse them.
5. In the caption of the phylogenetic tree, please write down the parameters of your analysis. Moreover, please use a sign like * for bootstrap values lower than 50.

Additional comments

Please revise your manuscript to make your study more reliable and well-written.

Reviewer 3 ·

Basic reporting

The article fails in several aspects, but the most important point is the use of the subtribe classification employed, which shows that the authors are not up-to-date with the work of the last decades and the recent publication of the tribe's circumscription.

It is necessary to include in the introduction the most recent history of the Cassiinae subtribe. The authors provide a description of the group and its circumscription up until 1982 (Irwin & Barneby, 1982). However, in recent decades, there is considerable evidence from various phylogenetic studies showing that the group is not monophyletic and that many of the characteristics that previously united the three genera (Cassia, Chamaecrista, and Senna) in older classifications are actually convergences, mainly considering Chamaecrista (see Bruneau et al. 2024, Phytokeys: https://phytokeys.pensoft.net/article/101716/). Currently, there is no longer a Cassiinae subtribe; instead, there is the Cassieae tribe, which comprises eight genera (https://phytokeys.pensoft.net/article/101716/).

Experimental design

The authors highlight that one of the important results of the study is the monophyly among the three groups. Again, this needs to be revised, as it is not supported by more recent studies. One possibility is the inclusion of the other four genera that are also part of the tribe, at least in the phylogenetic analysis with ITS.

The floral analyses are based on published studies, but I consider the approach to be quite superficial, lacking a clear methodology, especially in the case of Cassia and Chamaecrista.

Validity of the findings

The results of the study support the separation of the three genera and uphold the monophyly of each genus individually, which makes these findings quite valid.
There are some generalizations that should be avoided. For example, in the case of Chamaecrista, less than one-third of the genus has extrafloral nectaries, yet the authors characterize the genus as being typically associated with NEFs.

Additional comments

To avoid compromising the study, I suggest restructuring the work to clarify the issue of the new classification of the tribe. Even though the three genera are not closely related, they are the most representative in terms of species numbers within the Cassieae tribe, which makes this study quite interesting. Senna and Cassia are still considered sister groups, with only Chamaecrista apparently being more closely related to other genera.

---

## Round 0.2 · Minor Revisions

The authors have done a good job of addressing the reviewers' comments. I suggest the authors revise the paper to address reviewer three's (minor) comments before acceptance.

·

Basic reporting

The title of the manuscript is "Phylogeny of Cassieae Based on Seed Morphological and ITS evidence". I have evaluated this research paper properly. The authors have now updated the revised manuscript to incorporate all suggested changes. Finally, the manuscript might be accepted for publication.

Experimental design

Adequate, original primary research within the aim and scope of the journal.

Validity of the findings

Novel

Additional comments

The title of the manuscript is "Phylogeny of Cassieae Based on Seed Morphological and ITS evidence". I have evaluated this research paper properly. The authors have now updated the revised manuscript to incorporate all suggested changes. Finally, the manuscript might be accepted for publication.

Reviewer 2 ·

Basic reporting

Most of my previous suggestions, corrections and advices were applied to the ms by authors.

Experimental design

Most of my previous suggestions, corrections and advices were applied to the ms by authors.

Validity of the findings

Most of my previous suggestions, corrections and advices were applied to the ms by authors.

Additional comments

Most of my previous suggestions, corrections and advices were applied to the ms by authors.

Reviewer 3 ·

Basic reporting

The authors have revised this new submitted version, now incorporating the new delineation of the tribe that aligns more coherently with the theoretical and methodological context.There are minor errors that can be adjusted in the revision, such as the correct citation of the references used, both in the text and in the reference list. For example, the new circumscription of the tribe was proposed by Rando et al. (2024) (line 55 of manuscript), which should be cited as: Rando, J.G., Cota, M.M.T., Lima, A., Bortolozzi, R.L.C., Marazzi, B. and Conceição, A.S. 2024. Tribe Cassieae In: Bruneau, A. Queiroz, L.P., Ringelberg, J.J (Eds) Advances in Legume Systematics 14. Classification of Caesalpinioideae. Part 2: Higher-level classification. PhytoKeys 240: 83–102. https://doi.org/10.3897/phytokeys.240.101716. In the line 56 In line 56, "Cassina" is written instead of "Cassieae."

Experimental design

Including the new circumscription in the new methodology made perfect sense with the current classification, bringing more interesting and coherent results with the phylogenetic relationships accepted today.

Validity of the findings

Interesting results, not least because Asian species are less well known than those from the new world.

---

## Round 0.3 · accepted · Accept

I thank the authors for addressing the minor concerns of the reviewers.